# YKP-SLAM: A Visual SLAM Based on Static Probability Update Strategy for Dynamic Environments

**Lisang Liu [1,2]**, **Jiangfeng Guo [1,2],\* and Rongsheng Zhang [1,2]**

1   School of Electronic, Electrical Engineering and Physics, Fujian University of Technology,
    Fuzhou 350118, China
2   National Demonstration Center for Experimental Electronic Information and Electrical Technology Education,
    Fujian University of Technology, Fuzhou 350118, China
*   Correspondence: 2201905130@smail.fjut.edu.cn

**Abstract:** Visual simultaneous localization and mapping (SLAM) algorithms in dynamic scenes can incorrectly add moving feature points to the camera pose calculation, which leads to low accuracy and poor robustness of pose estimation. In this paper, we propose a visual SLAM algorithm based on object detection and static probability update strategy for dynamic scenes, named YKP-SLAM. Firstly, we use the YOLOv5 target detection algorithm and the improved K-means clustering algorithm to segment the image into static regions, suspicious static regions, and dynamic regions. Secondly, the static probability of feature points in each region is initialized and used as weights to solve for the initial camera pose. Then, we use the motion constraints and epipolar constraints to update the static probability of the feature points to solve the final pose of the camera. Finally, it is tested on the TUM RGB-D dataset. The results show that the YKP-SLAM algorithm proposed in this paper can effectively improve the pose estimation accuracy. Compared with the ORBSLAM2 algorithm, the absolute pose estimation accuracy is improved by 56.07% and 96.45% in low dynamic scenes and high dynamic scenes, respectively, and the best results are almost obtained compared with other advanced dynamic SLAM algorithms.

**Keywords:** Visual SLAM; dynamic scene; YOLOv5; K-means clustering; probability update

## 1. Introduction

Simultaneous localization and mapping (SLAM) is to estimate camera pose and build a map of the environment simultaneously during motion from sensor data collected by the robot. After decades of development, some very mature SLAM algorithms have emerged, such as PTAM [1], LSD-SLAM [2], DSO [3], ORB-SLAM2 [4], and VINS Mono [5], which are basically based on the assumption of static environments. However, in practical applications of robotics, motion scenes are more common than static scenes, and most application scenes encounter dynamic objects, e.g., pedestrians, vehicles, animals, etc. Dynamic objects can introduce anomalous "outliers" that disrupt the normal correspondence between image features, resulting in significant drift in camera pose. Some optimization algorithms, such as random sample consensus [6] (RANSAC) and graph optimization, can filter out a small number of weak dynamic features in the environment as outliers. These methods can achieve certain results for low-speed motion with a small number of outliers. Though, they are not able to process dynamic features very well for high-speed complex motion scenes, and the visual SLAM system might fail to track and localize. Therefore, it is particularly important to study SLAM algorithms in dynamic environments.

In order to solve the visual SLAM problem in a dynamic environment, the traditional method is to eliminate dynamic objects through geometric constraints and set a threshold according to the size of the reprojection error to distinguish static objects from dynamic objects. However, this method has two problems. (1) The method cannot distinguish the residuals caused by moving objects from those caused by mis-matching. (2) The

segmentation threshold is difficult to set; if the threshold set is too large, the static features will be mis-rejected, and if the segmentation threshold set is too small, it is difficult to completely reject the dynamic features in the environment. Therefore, the method is more suitable for a low dynamic environment. Additionally, in a high dynamic environment, the accuracy of dynamic feature detection is low, and the accuracy of pose estimation is poor.

In recent years, with the development of computer vision and deep learning, semantic constraints have been widely applied to visual SLAM problems in dynamic environments. The semantic constraint approach mainly applies semantic segmentation and target detection to obtain semantic information in the environment. By identifying and removing potential dynamic objects, the performance of visual SLAM in dynamic scenes can be greatly improved. The semantic segmentation algorithm can provide fine pixel-level object masks, but its real-time performance is poor. The improvement of segmentation accuracy and robustness often comes at the cost of huge computational cost. Even then, the segmentation boundary of an object cannot be very accurate. The target detection algorithm can quickly obtain the object frame of an object with low computational cost, but it cannot obtain accurate object boundaries, and if the features in the dynamic object frame are directly removed, it will lead to the false removal of some static features. Moreover, there are three problems with semantic constraints. (1) The actual motion is stationary, however, the algorithm cannot judge a semantic prior is a dynamic object or not, which may lead to the false removal of some static features. (2) It can only handle known objects labeled in the training set of the network but may still fail in the face of unknown moving objects, which leads to the missed detection of some dynamic features. (3) It deletes all dynamic features of semantic information discrimination and does not calculate the pose. This will lead to a reduction in constraints in pose calculation, knowing that dynamic features can still provide weak constraints for pose calculation. If it is deleted directly, it will lead to a decrease in the accuracy of pose estimation.

To address the above problems, in order to improve the pose estimation accuracy and robustness of the SLAM system in a dynamic environment, this paper proposes a YKP-SLAM algorithm in a dynamic environment. On the basis of ORBSLAM2, YKP-SLAM adds three major processes: YOLOv5 target detection, improved K-means clustering, and probability updating strategy. Our experiments prove that the YKP-SLAM algorithm can effectively reduce the tracking error and improve the accuracy and robustness of the SLAM system, both in a slow-moving dynamic environment and in a fast-moving dynamic environment.

The main contributions of this paper are as follows:

(1) We incorporate the lightweight YOLOv5 object detection algorithm into the SLAM system, which can quickly and accurately provide accurate semantic priors for subsequent operations.

(2) A K-means clustering algorithm specifically for depth images is proposed, which can select the number of clusters adaptively and can segment dynamic object contours from dynamic object frames quickly and accurately.

(3) A method for initializing static probability is proposed. The image is divided into three regions by combining YOLOv5 and improved K-means clustering. Then, the initial poses are solved by probability initialization of feature points in each region separately. More accurate initial poses are provided for the subsequent motion constraints and polar constraints.

(4) A probability update strategy based on motion constraints and epipolar constraints is proposed. Probability updates are performed for all feature points in the image. Then, all feature points are added to the pose calculation to solve the final pose.

## 2. Related Work

### 2.1. Dynamic SLAM Based on Traditional Method

Traditional dynamic SLAM algorithms are mainly based on geometric constraints to filter out dynamic feature points in the environment. For example, Zou [7] et al. project

feature points from the previous frame onto the current frame and calculate the 2D repro-jection error of matching points with the current frame and classify feature points into static and dynamic feature points according to the magnitude of the reprojection error. Wang [8] et al. detected the matched outlier points in two adjacent frames by epipolar constraint and then fused the clustering information of the depth map provided by RGB-D cameras to identify the moving targets in the scene. Dai [9] et al. proposed a static object geometry prior method in a feature-based SLAM framework. The algorithm utilizes the connectivity of map points to separate moving objects from the static background, thus reducing the impact of moving objects on the pose estimation.

In addition to geometric constraints, optical flow methods are also used to distinguish dynamic and static features. For example, Klappstein [10] et al. defined the likelihood of "moving objects in the scene" based on the motion metric calculated by optical flow. Fang [11] et al. improved the optical flow method to detect dynamic targets based on point matching techniques and uniform sampling strategies and introduced a Kalman filter to enhance detection and tracking. FlowFusion [12] estimated the optical flow of two adjacent frames through a PWC-Net [13] network, and at the same time, estimated the camera pose based on the intensity and depth of the two adjacent frames and then used the estimated optical flow and camera motion to compute the 2D scene flow and finally used the 2D scene flow for dynamic feature segmentation.

### 2.2. Dynamic SLAM Based on Semantic Constraints

In recent years, deep-learning-based image semantic segmentation and target recogni-tion have been widely used, and the detection methods have evolved greatly in terms of efficiency and accuracy. Many researchers have tried to solve the dynamic SLAM problem by removing potential dynamic objects through semantic tagging or target detection prepro-cessing. For example, Yang [14] et al. used the target detection network Faster R-CNN [15] to detect dynamic objects and then performed geometric matching with the current frame and keyframes to determine whether they are dynamic objects. Yu [16] et al. proposed the DS-SLAM algorithm, combining a semantic segmentation network and optical flow method to provide a semantic representation of octree maps, thus reducing the dynamic objects. The DynaSLAM proposed by Bescos [17] et al. uses a combination of multi-view geometry and Mask RCNN [18] to detect and filter dynamic targets. ZHANG Jinfeng [19] et al. used the target detection network YOLOv3 [20] to filter dynamic feature points in the scene, which effectively reduced the trajectory error of the SLAM system. Zhong [21] et al. proposed Detect-SLAM combined with the target detection network SSD [22] to identify dynamic targets, such as pedestrians and vehicles, in the environment as a priori dynamic targets and then filter the feature points on the a priori dynamic target to improve its localization accuracy. Blitz-SLAM [23] obtains the mask of the object by BlitzNet [24], then completes the mask by depth information, and finally classifies the static feature points and dynamic feature points by epipolar constraints.

## 3. Materials and Methods

### 3.1. System Architecture

The algorithm framework of YKP-SLAM is shown in Figure 1. Based on ORBSALM2, we added the YOLOv5 target detection algorithm and the improved K-means clustering algorithm to the fore-end and added a complete probability update strategy to the back-end pose calculation. The algorithmic flow of YKP-SLAM can be described as follows. Firstly, the RGB image is detected by YOLOv5 target detection algorithm to obtain the dynamic object frame, and at the same time, the ORB [25] feature points are extracted from the RGB image. Secondly, the depth values of the pixel points are clustered within the dynamic object frame by the improved K-means clustering algorithm combined with the depth image. The results of YOLOv5 target detection and K-means clustering are used to segment the image into static regions, suspicious static regions, and dynamic regions, initialize the static probability of feature points within each region, and add them as weights

to the camera pose estimation to calculate the initial camera pose $T_{cw1}$. Finally, the static probability of feature points is updated by the motion constraint and the epipolar constraint, and the second stage pose $T_{cw2}$ and the final pose $T_{cw}$ of the camera are solved, respectively.

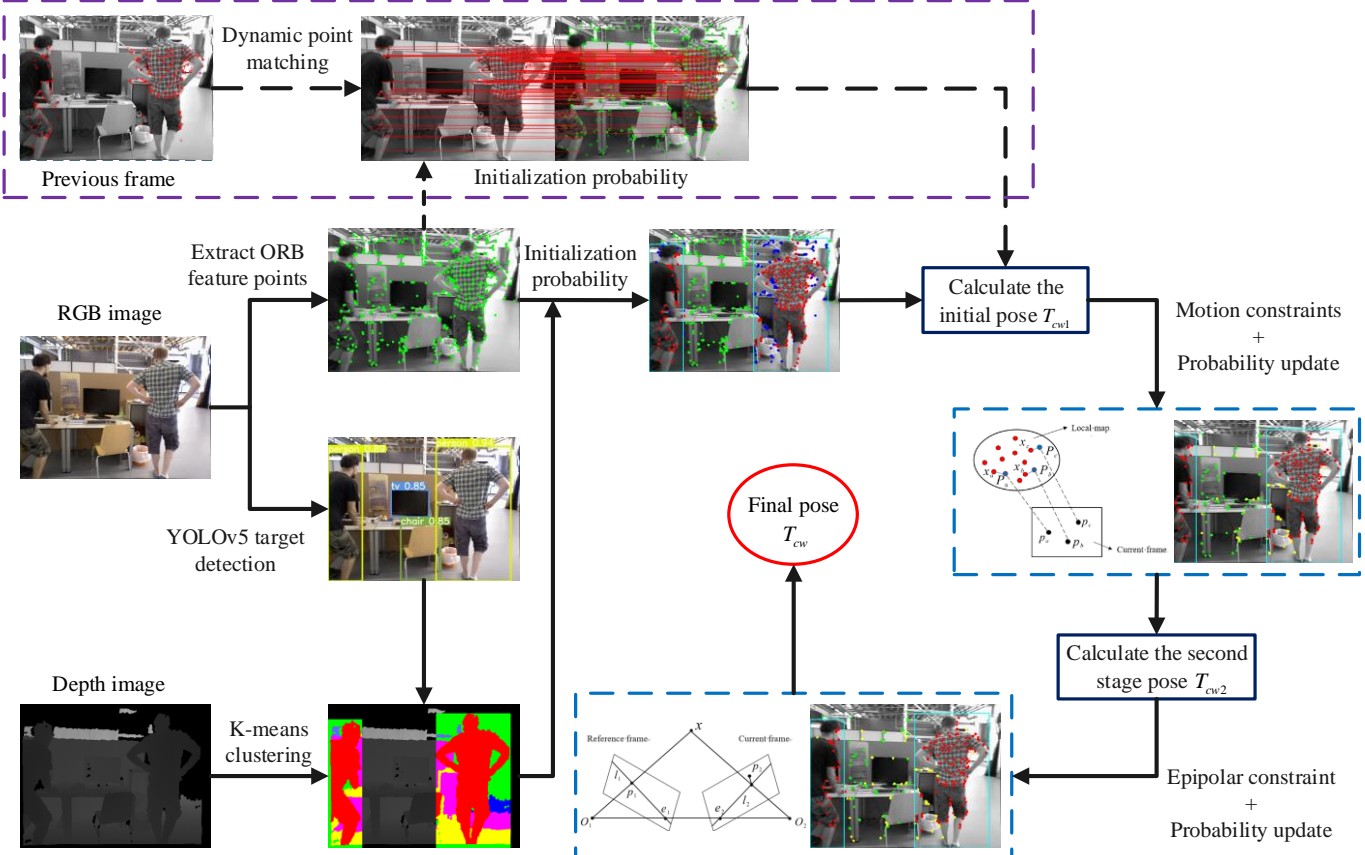

**Figure 1.** The algorithmic framework of YKP-SLAM. In the image, green points represent static points, blue points represent suspicious static points, red points represent dynamic points, and yellow points represent points where the probability changes.

Of course, we also considered the failure of the YOLOv5 algorithm. When YOLOv5 fails, the dynamic object frame cannot be obtained. Then, at this time, we perform feature matching between the feature points in the current frame and the dynamic feature points in the previous frame. Mark the successfully matched feature points of the current frame as dynamic feature points, and mark the remaining feature points as static feature points. The only difference from a normal operation is that the characteristic points are divided into three categories in a normal operation, and the characteristic points in a fault operation are divided into two categories. The subsequent static probability initialization method and probability update strategy are the same. The feature point classification process of YOLOv5 fault runtime is shown in the purple dashed box in Figure 1.

### 3.2. YOLOv5 Target Detection

You Only Look Once (YOLO) is a regression-based target detection algorithm. It is the pioneering work of the one-stage method. It was released by Ultralytics on 10 June 2020. It is one of the most widely used target detection algorithms. It solves target detection as a regression problem and directly obtains the bounding box position and classification of the predicted object from an input image. It ensures the accuracy while taking into account the real-time performance and achieves very good speed and accuracy. YOLOv5 proposes a total of 4 network models: YOLOv5s, YOLOv5m, YOLOv5l, and YOLOv5x. The network structure of the four models is the same; the difference is that the depth_multiple

and width_multiple parameters can be used to control the depth of the model and the number of convolution kernels, respectively. Among them, YOLOv5s is the network with the smallest network depth and the smallest feature map width. It occupies only 7.5 M of memory. Its detection speed on TeslaP100 reaches 140FPS, which fully meets real-time performance. The other three are continuously deepened and widened on this basis, with improved accuracy and slower speed.

In order to meet the real-time nature of the SLAM system, the fastest YOLOv5s algorithm is adopted, which is embedded in the fore-end of the SLAM system, to perform target detection on each RGB image passed by the camera and obtain the bounding box position of the object and its category. In the bounding box, the people and animals are located as dynamic object boxes $DB$. The target detection results of YOLOv5s are shown Figure 2. The yellow frame in Figure 2 is the dynamic object box. It can be seen from the figure that whether the person is on the front, side, back, or only half of the body is exposed, YOLOv5 can be accurately framed.

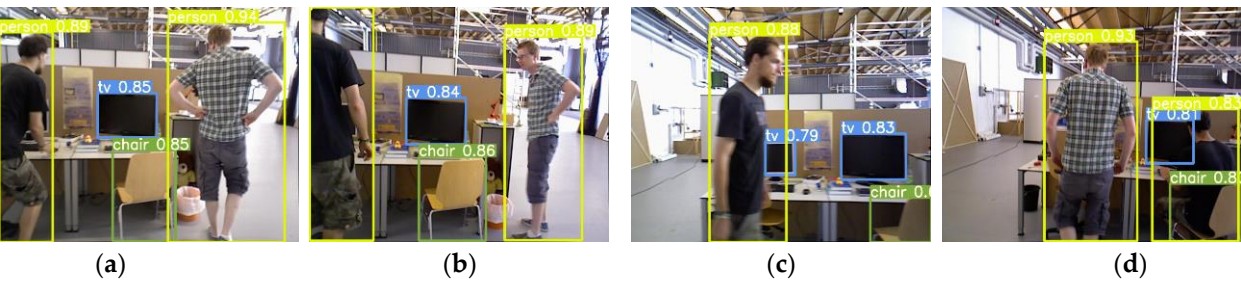

|           |           |           |           |
|:---------:|:---------:|:---------:|:---------:|
| (**a**)   | (**b**)   | (**c**)   | (**d**)   |

**Figure 2.** YOLOv5 target detection results. (**a**–**d**) represents the detection results of the YOLOv5 algorithm in several different scenes.

### 3.3. Improved Adaptive K-means Clustering Algorithm

Although the YOLOv5 target detection algorithm can quickly and accurately locate the bounding boxes of dynamic objects, it cannot obtain an accurate dynamic object mask. Therefore, this paper proposes an adaptive K-means clustering segmentation algorithm based on depth images, which can segment dynamic objects from the dynamic object box $DB$ quickly and accurately.

The K-means algorithm is an unsupervised clustering algorithm, which is easy to implement and runs fast. However, the traditional K-means clustering algorithm pre-specifies the number of clusters and randomly initializes the cluster centers according to experience, which is likely to cause too many iterations of the algorithm or misclassification. Since the number of clusters is artificially set in advance, the direct application of the traditional K-means clustering algorithm to depth image clustering will have the following two problems:

(1) If the number of clusters set is too large, a complete dynamic objects would be divided into multiple categories, which might cause incomplete segmentation of dynamic objects.

(2) If the number of clusters set is too small, the dynamic objects cannot be separated from the static background.

In order to solve the above problems, an improved adaptive K-means algorithm is proposed in this paper. The algorithm can automatically generate the optimal number of clusters and the initial cluster centers, so that dynamic objects can be segmented from the static background more quickly and accurately. The steps of the improved K-means algorithm are as follows:

(1) Take out the depth image $IDB_i$ in the dynamic object frame $DB$ and count the total number of pixels $M$ and the maximum pixel depth $D_{max}$ in $IDB_i$.

(2) Solve the histogram of the depth image $IDB_i$ and divide the data of the histogram into $k$ segments:

$$k = \frac{D_{\max}}{T} \tag{1}$$

where $T$ is the segmentation threshold, whose size can determine the number of clusters. Since the depths of dynamic objects do not change much in the two adjacent frames, we first use the depth mean $D_p$ of dynamic feature points in the previous frame as the prior of the depth value of dynamic objects in the current frame. Then, the ratio $\lambda$ of the number of pixels in the dynamic object in the previous frame to the number of pixels in the dynamic object frame is calculated. Finally, find the neighborhood $U(D_p, \delta) = \{x \mid D_p - \delta < x < D_p + \delta\}$ of point $D_p$ in the histogram of the depth image $IDB_i$, so that the number of pixels in the neighborhood is equal to $\lambda M$; then, the size of the segmentation threshold $T$ is the range of the neighborhood.

$$T = 2\delta \tag{2}$$

(3) We take $k$ as the number of clusters for subsequent K-means clustering and take the maximum depth value of each piece of data as the initial cluster center for each category.

(4) The K-means clustering algorithm obtains a depth image segmentation graph based on the number of clusters calculated in step (3) and the initial cluster centers.

Since the depth values of dynamic objects do not change too much within the two adjacent frames, the depth mean value $D_p$ of dynamic features in the previous frame is used as a criterion, and then, the pixel depth mean value of each cluster in the dynamic object box is solved, and the cluster with a pixel depth mean value closest to the depth mean value of dynamic points in the previous frame is marked as a dynamic region; the other clusters in the dynamic object box are marked as suspicious static regions, and the regions outside the dynamic object box are marked as static regions. The whole dynamic region classification process is shown in Figure 3.

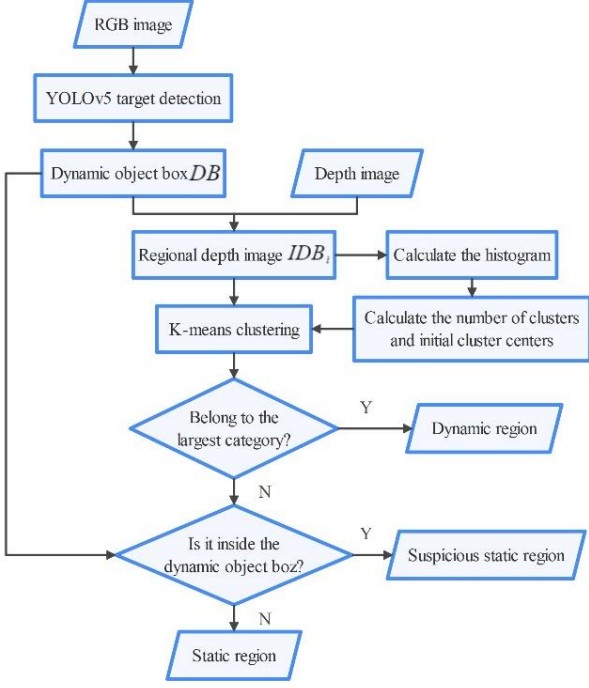

**Figure 3.** Schematic diagram of dynamic region division.

The results of K-means clustering are shown in Figure 4. From the figure, we can see that the improved K-means clustering algorithm proposed in this paper can segment people from the background completely and does not lead to mis-segmentation.

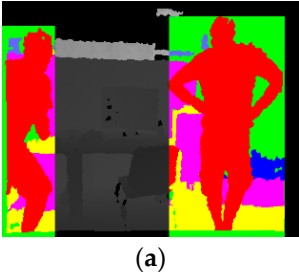 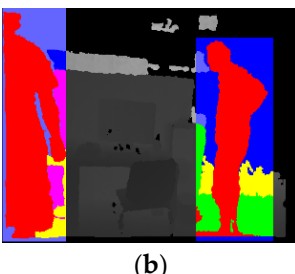 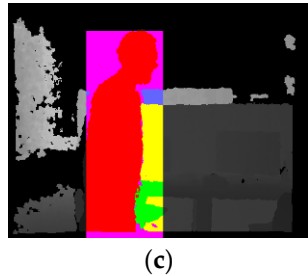 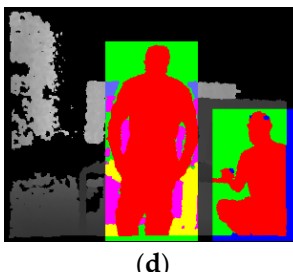

| (a) | (b) | (c) | (d) |

**Figure 4.** Improved K-means clustering, where each color represents one class. The red region is the dynamic region, and the other colored regions are suspicious static regions. (**a**–**d**) represents the clustering results of the improved K-means clustering algorithm in several different scenes.

*3.4. Initialize the Static Probability and Calculate the Initial Camera Pose*

In this paper, the YOLOv5 target detection algorithm and the improved adaptive K-means clustering algorithm are used to segment the image into dynamic regions, suspicious static regions, and static regions. In order to obtain a more accurate initial pose, the feature points in different regions are assigned static probability initial values of

$$
Static\ probability \begin{cases} \omega_a = 0 & Dynamic\ region \\ \omega_b = 0.5 & Suspicious\ static\ region \\ \omega_c = 1 & Static\ region \end{cases} \tag{3}
$$

These initial static probabilities are then used as weights for the pose calculation, and the initial pose $T_{cw1}$ for the current frame is calculated according to the weighted minimization reprojection error.

The structure of the camera pose $T_{cw1}$ is

$$
SE(3) = \left\{ T_{cw1} = \begin{bmatrix} R_{cw1} & t_{cw1} \\ 0^T & 1 \end{bmatrix} \in \mathbb{R}^{4 \times 4} \mid R_{cw1} \in SO(3), t_{cw1} \in \mathbb{R}^3 \right\} \tag{4}
$$

where $R_{cw1}$ is the rotation matrix, and $t_{cw1}$ is the translation vector.

$T_{cw1}$ can be solved by Equation (5).

$$
\begin{aligned}
T_{cw1} = \mathrm{argmin}( &\sum_{a=1}^{N_a} \|KT_{cw1}x_a - p_a\|_{\Sigma_1}^2 + \\
&\sum_{b=1}^{N_b} \|KT_{cw1}x_b - p_b\|_{\Sigma_2}^2 + \\
&\sum_{c=1}^{N_c} \|KT_{cw1}x_c - p_c\|_{\Sigma_3}^2 )
\end{aligned} \tag{5}
$$

Among them

$$
\begin{aligned}
\Sigma_1 &= \omega_a \times n \times E \\
\Sigma_2 &= \omega_b \times n \times E \\
\Sigma_3 &= \omega_c \times n \times E
\end{aligned} \tag{6}
$$

Where, $p_a, p_b, p_c$ are the 2D pixel point coordinates of dynamic feature points, suspicious static points, and static points in the current frame, respectively, while $x_a, x_b, x_c$ are the coordinates of their corresponding matching 3D map points. $\Sigma_1, \Sigma_2, \Sigma_3$ is the information matrix of feature points in each region, $n$ is the number of layers of the image pyramid where the current feature point is located, and $E$ is the unit matrix of $3 \times 3$. $N_a, N_b, N_c$ are the numbers of dynamic feature points, suspicious static points, and static points in the current frame, respectively.

### 3.5. Probability Update Based on Motion Constraints

The traditional geometric method distinguishes dynamic points and static points by the size of the reprojection error and sets the threshold value and judges the points with a reprojection error larger than the threshold value as dynamic points and those smaller than the threshold value as static points. The threshold size of this method is difficult to set, which can easily lead to mis-segmentation of dynamic and static points. Therefore, this paper proposes a new segmentation method that uses the motion distance of the a priori dynamic point $p_a$ judged by the front-end of the SLAM system (YOLOv5 and K-means) as a scale to update the static probability of the suspicious static point $p_b$ and static point $p_c$. The schematic diagram of the motion constraint is shown in Figure 5.

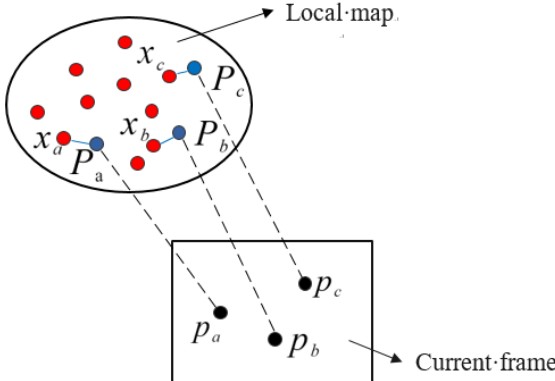

**Figure 5.** Schematic diagram of motion constraints, where the ellipse represents the local map, the rectangle represents the current frame, the red point inside the ellipse represents the local map point, the blue point represents the 3D point of the current frame feature point back-projected to the world coordinate system, and the line between the red point and the blue point represents the motion distance of the feature point.

We now know the initial pose $T_{cw1}$ and the camera internal reference $K$ of the current frame, and we can also directly obtain the depth information $Z$ of the feature points through the depth camera. Then, we first back-project the dynamic point $p_a$ in the current frame to the world coordinate system to obtain the 3D point coordinate $P_a$ in the world coordinate system.

$$P_a = \begin{bmatrix} X_a \\ Y_a \\ Z_a \end{bmatrix} = T_{wc1} K p_a \tag{7}$$

Calculate the square value $L_a$ of the movement distance between the back-projection point $P_a$ and the corresponding map point $x_a$:

$$L_a = (X_a - X_a')^2 + (Y_a - Y_a')^2 + (Z_a - Z_a')^2 \tag{8}$$

where $\begin{bmatrix} X_a' & Y_a' & Z_a' \end{bmatrix}^T$ are the 3D point coordinates of the map point $x_a$.

Similarly, the squares of the motion distances of the suspicious static point $p_b$ and the static point $p_c$ can be solved as $L_b$ and $L_c$, respectively.

Then, solve the mean $\mu_L$ and variance $S_L$ of the square of the motion distance of the dynamic point $p_a$ in the current frame:

$$\mu_L = \frac{\sum\limits_{a=1}^{N_a} L_a}{N_a} \tag{9}$$

$$S_L = \sqrt{\frac{\sum\limits_{a=1}^{N_a}(L_a - \mu_L)^2}{N_a}} \tag{10}$$

By comparing the motion distance of the suspicious static point $p_b$, static point $p_c$, and dynamic point $p_a$ to update their static probability, this paper designs a sigmoid function to calculate the static probability of each suspicious static point $p_b$ and static point $p_c$ as follows:

$$\omega_{b1} = \frac{1}{1 + \exp(\alpha(\frac{L_b - \mu_L}{S_L}))} \tag{11}$$

$$\omega_{c1} = \frac{1}{1 + \exp(\alpha(\frac{L_c - \mu_L}{S_L}))} \tag{12}$$

where $\alpha$ is a coefficient greater than 0.

Update the static probability of each feature point in each region in combination with the initial static probability:

$$\begin{aligned} \omega_a &= \omega_a \\ \omega_b &= \omega_b \times \omega_{b1} \\ \omega_c &= \omega_c \times \omega_{c1} \end{aligned} \tag{13}$$

Based on the updated static probability of the feature points, the static probabilities are brought into Equation (5) to calculate the camera pose $T_{cw2}$ in the second stage.

### 3.6. Probability Update Based on Epipolar Constraint

As shown in Figure 6, $O_1, O_2$ is the camera optical center at the moment of the current frame and reference frame, respectively, and $p_1, p_2$ is a pair of matching points between the current frame and reference frame. $x$ is the map point corresponding to the $p_1$ point on the reference frame, and the projection point of this point on the current frame should be located on the polar line $l_2$ if the point is stationary, or not on the polar line if it is moving. In this paper, the static probability of the feature points is updated based on the distance from point $p_2$ to the polar line $l_2$.

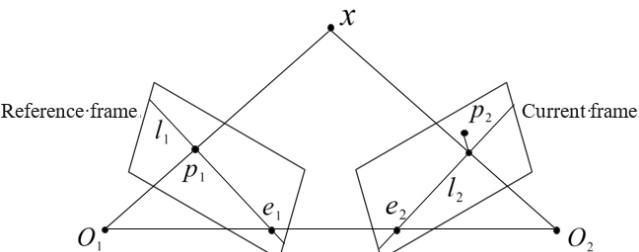

**Figure 6.** Schematic diagram of epipolar constraint.

Through the current frame camera pose $T_{cw2}$ and the reference frame camera pose $T_{cwr}$ solved in the second stage, the rotation matrix and translation matrix $t_{2r}$ between the two frames can be solved:

$$R_{2r} = R_{cw2} \times R_{cwr}^{-1} \tag{14}$$

$$t_{2r} = -R_{cw2} \times R_{cwr}^{-1} \times t_{cwr} + t_{cw2} \tag{15}$$

Among them, $R_{cw2}$ and $t_{cw2}$ are the rotation matrix and translation matrix of the current frame, respectively, and $R_{cwr}$ and $t_{cwr}$ are the rotation matrix and translation matrix of the reference frame.

Fundamental matrix $F$

$$F = K^{-T}(t_{2r})^{\wedge}R_{2r}K^{-1} \tag{16}$$

Solve the polar equation corresponding to the feature point on the reference frame to the current frame according to the fundamental matrix. The polar equation is expressed as

$$\begin{bmatrix} A & B & C \end{bmatrix}^T = F \begin{bmatrix} u_1 & v_1 & 1 \end{bmatrix} \tag{17}$$

$\begin{bmatrix} u_1 & v_1 & 1 \end{bmatrix}$ is the homogeneous coordinate of the reference frame feature point $p_1$.

Calculate the square of the polar distance from the feature point of the current frame to the corresponding polar line:

$$H = \frac{(Au_2 + Bv_2 + C)^2}{A^2 + B^2} \tag{18}$$

$\begin{bmatrix} u_2 & v_2 & 1 \end{bmatrix}$ is the homogeneous coordinate of the current frame feature point $p_2$.

According to the above Equations (16)–(18), the polar distance $H_a, H_b, H_c$ of the dynamic point, suspicious static point, and static point of the current frame can be calculated, respectively.

Calculate the mean $\mu_H$ and variance $S_H$ of the polar distance of the dynamic points, as with the motion constraints:

$$\mu_H = \frac{\sum\limits_{a=1}^{N_a} H_a}{N_a} \tag{19}$$

$$S_H = \sqrt{\frac{\sum\limits_{a=1}^{N_a} (H_a - \mu_H)^2}{N_a}} \tag{20}$$

By comparing the polar distance of the suspicious static point $p_b$, static point $p_c$, and dynamic point $p_a$ to update their static probability

$$\omega_{b2} = \frac{1}{1 + \exp(\beta(\frac{H_b - \mu_H}{S_H}))} \tag{21}$$

$$\omega_{c2} = \frac{1}{1 + \exp(\beta(\frac{H_c - \mu_H}{S_H}))} \tag{22}$$

where $\beta$ is a coefficient greater than 0.

Update the final static probability of the feature points in each region using the static probability of epipolar constraints:

$$\begin{aligned} \omega_a &= \omega_a \\ \omega_b &= \omega_b \times \omega_{b2} \\ \omega_c &= \omega_c \times \omega_{c2} \end{aligned} \tag{23}$$

The final camera pose $T_{cw}$ can be calculated from the final static probability of the feature points and Equation (5).

## 4. Experiments and Analysis

In order to evaluate the performance of the YKP-SLAM algorithm, this paper uses the public TUM RGB-D dataset [26] to conduct the experiments. The TUM dataset is produced by the University of Munich, Germany, and uses a Kinect sensor to capture information at a rate of 30 HZ with an image resolution of 640 ∗ 480 and uses a high-precision motion capture system VICON with an inertial measurement system while acquiring image data. The camera position and pose data are acquired in real time, which can be approximated as the real positional data of the RGB-D camera. In this paper, we mainly use eight dynamic scene sequences from the TUM RGB-D dataset for experiments, which are divided into two categories: walking and sitting. The sitting dataset series are low dynamic scenes, in

which two people are sitting in front of a table and chatting, with low motion. The walking dataset series are high dynamic scenes, in which two people are walking in front of or around a table, with high motion. For each type of dataset series, the camera motion is also divided into four states, where static means the camera is at rest, xyz means the camera is moving along the spatial X-Y-Z axis in translation, rpy means the camera is rotating in a flip angle, pitch angle, and yaw angle, and halfsphere means the camera is moving along the trajectory of a hemisphere with a diameter of 1 m.

The experiments were run on a server with Ubuntu 18.04, a GeForce RTX 3060 graphics card with 12 GB of video memory, a 7-core Intel(R) Xeon(R) CPU, and 20 GB of RAM.

### 4.1. Comparison with ORBSLAM2

Since the YKP-SLAM algorithm proposed in this paper is improved on the basis of ORBSLAM2, a comparison experiment with ORBSLAM2 is conducted first. In this paper, the absolute trajectory error (ATE) and relative pose error (RPE) [26] are adopted to evaluate algorithm accuracy. The absolute trajectory error is the direct difference between the estimated and real poses, which can reflect the algorithm accuracy and global consistency of the trajectory very intuitively. The relative trajectory error contains the relative translation error and relative rotation error, which are directly measured by the odometer. The experimental results are shown in Tables 1 and 2, where RMSE denotes the root mean square error, Mean denotes the mean error, and Std denotes the standard deviation.

**Table 1.** Comparison of absolute trajectory error (ATE) between ORB-SLAM2 and YKP-SLAM.

| Sequences | ORB-SLAM2/m | | | YKP-SLAM/m | | | Improvement/% | | |
|---|---|---|---|---|---|---|---|---|---|
| | RMSE | Mean | Std | RMSE | Mean | Std | RMSE | Mean | Std |
| sitting_xyz | 0.0111 | 0.0093 | 0.0059 | 0.0072 | 0.0065 | 0.0033 | 35.14 | 30.11 | 44.07 |
| sitting_half | 0.0437 | 0.0360 | 0.0247 | 0.0153 | 0.0132 | 0.0076 | 64.99 | 63.33 | 69.23 |
| sitting_static | 0.0128 | 0.0120 | 0.0046 | 0.0052 | 0.0043 | 0.0028 | 59.38 | 64.17 | 39.13 |
| sitting_rpy | 0.0358 | 0.0293 | 0.0205 | 0.0268 | 0.0237 | 0.0126 | 25.13 | 19.11 | 38.53 |
| walking_xyz | 0.5185 | 0.4420 | 0.2711 | 0.0147 | 0.0130 | 0.0068 | 97.16 | 97.06 | 97.49 |
| walking_half | 0.5820 | 0.4571 | 0.3603 | 0.0245 | 0.0220 | 0.0107 | 95.79 | 95.19 | 97.03 |
| walking_static | 0.2742 | 0.2286 | 0.1514 | 0.0063 | 0.0056 | 0.0026 | 97.70 | 97.55 | 98.28 |
| walking_rpy | 1.5320 | 1.4262 | 0.5594 | 0.0702 | 0.0489 | 0.0514 | 95.41 | 96.57 | 90.81 |

**Table 2.** Comparison of relative pose error (RPE) between ORB-SLAM2 and YKP-SLAM.

| Sequences | ORB-SLAM2/m | | | YKP-SLAM/m | | | Improvement/% | | |
|---|---|---|---|---|---|---|---|---|---|
| | RMSE | Mean | Std | RMSE | Mean | Std | RMSE | Mean | Std |
| sitting_xyz | 0.0148 | 0.0126 | 0.0077 | 0.0079 | 0.0070 | 0.0038 | 46.62 | 44.44 | 50.65 |
| sitting_half | 0.0227 | 0.0121 | 0.0192 | 0.0137 | 0.0108 | 0.0084 | 39.64 | 10.74 | 56.25 |
| sitting_static | 0.0180 | 0.0169 | 0.0063 | 0.0058 | 0.0055 | 0.0031 | 67.78 | 67.46 | 50.79 |
| sitting_rpy | 0.0256 | 0.0208 | 0.0148 | 0.0232 | 0.0171 | 0.0151 | 9.38 | 17.79 | −2.27 |
| walking_xyz | 0.0382 | 0.0303 | 0.0233 | 0.0139 | 0.0116 | 0.0076 | 63.61 | 61.72 | 67.38 |
| walking_half | 0.0452 | 0.0317 | 0.0322 | 0.0196 | 0.0148 | 0.0128 | 56.64 | 53.31 | 60.25 |
| walking_static | 0.0473 | 0.0291 | 0.0373 | 0.0072 | 0.0062 | 0.0031 | 84.78 | 78.69 | 91.69 |
| walking_rpy | 0.0429 | 0.0316 | 0.0291 | 0.0317 | 0.0218 | 0.0239 | 26.11 | 31.01 | 17.97 |

The improvement rates in the table are calculated as follows:

$$\eta = \left(1 - \frac{\beta}{\alpha}\right) \times 100\% \tag{24}$$

where $\eta$ represents the algorithm improvement rate, $\beta$ represents the experimental results of the YKP-SLAM algorithm, and $\alpha$ represents the experimental results of the ORBSLAM2 algorithm.

Tables 1 and 2 show the quantitative evaluation of the errors, from which it can be seen that in the low dynamic scene sitting dataset series, the average improvement of the RMSE of absolute and relative trajectory errors of the YKP-SLAM algorithm compared with the ORBSLAM2 algorithm is 46.16% and 40.86%, respectively. The average improvement of the RMSE of absolute and relative trajectory errors of this algorithm over ORBSLAM2 is 96.52% and 57.79%, respectively, in the walking data set series of high dynamic scenes, which shows that the YKP-SLAM algorithm has a great improvement over the traditional ORBSLAM2 algorithm in both low and high dynamic scenes. The trajectory accuracy is greatly improved in both low and high dynamic scenes.

Figures 7 and 8 show the absolute trajectory error distributions of the ORBSLAM2 algorithm and the YKP-SLAM algorithm under the low dynamic sequences s_xyz, s_half and the high dynamic sequences w_xyz, w_half, respectively. Figures 9 and 10 show the comparison of the estimated trajectory and the real trajectory of the ORBSLAM2 algorithm and the YKP-SLAM algorithm under the low dynamic sequences s_xyz, s_half and the high dynamic sequences w_xyz, w_half, respectively. It can be seen that under the low dynamic sequences s_xyz and s_half, the absolute trajectory error of the YKP-SLAM algorithm is slightly smaller than that of the ORBSLAM2 algorithm, and the estimated trajectory is closer to the real trajectory than the ORBSLAM2 algorithm. Under the high dynamic sequences w_xyz and w_half, the absolute pose error of the YKP-SLAM algorithm is smaller than that of the ORBSLAM2 algorithm, and the estimated trajectory is still very close to the real trajectory, while the estimated trajectory of the ORBSLAM2 algorithm is far away from the real trajectory. This proves that the YKP-SLAM algorithm can effectively improve the pose estimation accuracy of the SLAM system in low dynamic and high dynamic scenes.

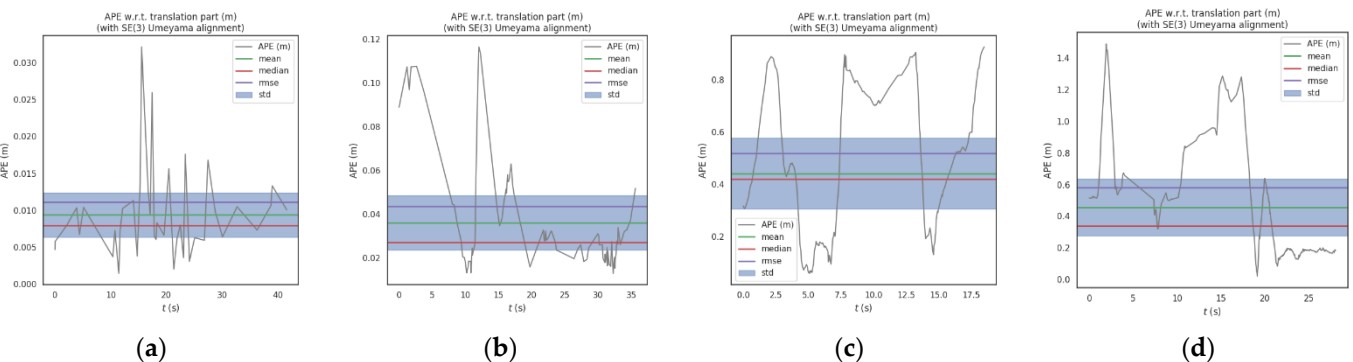

**Figure 7.** Absolute trajectory error distribution of ORBSLAM2 algorithm. (**a**) s_xyz. (**b**) s_half. (**c**) w_xyz. (**d**) w_half.

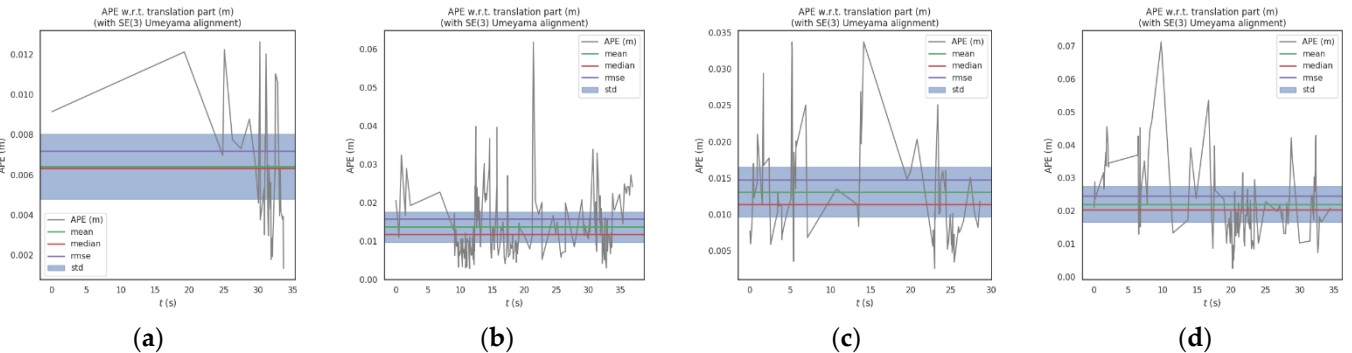

**Figure 8.** Absolute trajectory error distribution of YKP-SLAM algorithm. (**a**) s_xyz. (**b**) s_half. (**c**) w_xyz. (**d**) w_half.

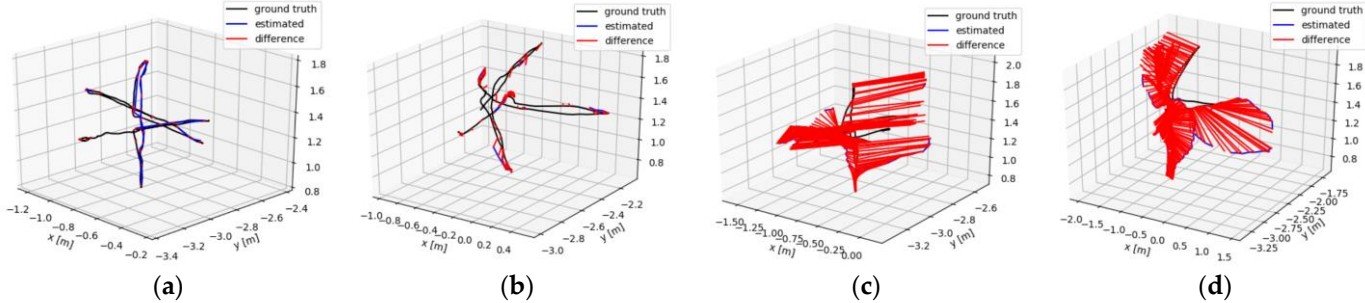

**Figure 9.** Comparison of estimated trajectory and real trajectory of ORBSLAM2 algorithm. The colored line is the estimated trajectory, and the gray line is the real trajectory. (**a**) s_xyz. (**b**) s_half. (**c**) w_xyz. (**d**) w_half.

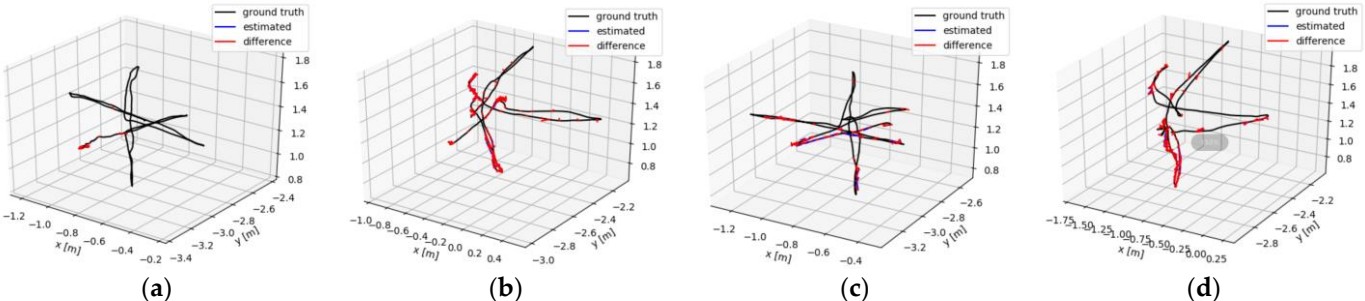

**Figure 10.** Comparison of estimated trajectory and real trajectory of YKP-SLAM algorithm. The colored line is the estimated trajectory, and the gray line is the real trajectory. (**a**) s_xyz. (**b**) s_half. (**c**) w_xyz. (**d**) w_half.

### 4.2. Comparison with Advanced Dynamic SLAM Algorithms

In order to verify the superiority of the YKP-SLAM algorithm, DS-SLAM [16], DynaSLAM [17], and Blitz-SLAM [23] are selected for comparison experiments with YKP-SLAM in this paper. The root mean square error RMSE and variance Std in the absolute trajectory error are selected as the evaluation metrics for verification. The experimental results are shown in Table 3, where the bold font indicates the best results. Among them, the DS-SLAM and DynaSLAM codes were open sourced as well as the experimental data, while the Blitz-SLAM algorithm code was not open sourced. As can be seen from the table, the YKP-SLAM algorithm achieves almost the best results compared to the other dynamic SLAM algorithms, both in high dynamic scenes and in low dynamic scenes. The performance is slightly worse under the s_rpy and w_rpy data sets, which is caused by the fact that the camera motion is too large at this time, making the YOLOv5 target detection results less accurate.

**Table 3.** Comparison of absolute trajectory error (ATE) between YKP-SLAM algorithm and other dynamic SLAM algorithms.

| Sequences | DS-SLAM/m | | DynaSLAM/m | | Blitz-SLAM/m | | YKP-SLAM/m | |
|---|---|---|---|---|---|---|---|---|
| | RMSE | Std | RMSE | Std | RMSE | Std | RMSE | Std |
| sitting_xyz | 0.0187 | 0.0119 | 0.0135 | 0.0063 | 0.0148 | 0.0069 | **0.0072** | **0.0033** |
| sitting_half | 0.0162 | **0.0061** | 0.0193 | 0.0084 | 0.0160 | 0.0076 | **0.0153** | 0.0076 |
| sitting_static | 0.0065 | 0.0033 | 0.0085 | 0.0051 | / | / | **0.0052** | **0.0028** |
| sitting_rpy | **0.0266** | 0.0153 | 0.0865 | 0.0516 | / | / | 0.0268 | **0.0126** |
| walking_xyz | 0.0247 | 0.0186 | 0.0176 | 0.0086 | 0.0153 | 0.0078 | **0.0147** | **0.0068** |
| walking_half | 0.0303 | 0.0159 | 0.0273 | 0.0130 | 0.0256 | 0.0126 | **0.0245** | **0.0107** |
| walking_static | 0.0081 | 0.0036 | 0.0067 | 0.0031 | 0.0102 | 0.0052 | **0.0063** | **0.0026** |
| walking_rpy | 0.4442 | 0.2350 | 0.0389 | 0.0237 | **0.0356** | **0.0220** | 0.0702 | 0.0514 |

### 4.3. Ablation Experiment

In order to verify the effectiveness of the improved K-means clustering algorithm and probability update strategy proposed in this paper, we conduct ablation experiments, and the experimental results are shown in Table 4 The bold font indicates the best results, and the underlined ones represent the second best results.

**Table 4.** Comparison of absolute trajectory error of ablation experiment.

| Sequences | Y-SLAM/m | | YK-SLAM/m | | YKP-SLAM/m | |
|---|---|---|---|---|---|---|
| | RMSE | Std | RMSE | Std | RMSE | Std |
| sitting_xyz | 0.0168 | 0.0079 | <u>0.0129</u> | <u>0.0068</u> | **0.0072** | **0.0033** |
| sitting_half | 0.0858 | 0.0178 | <u>0.0189</u> | <u>0.0084</u> | **0.0153** | **0.0076** |
| sitting_static | <u>0.0072</u> | <u>0.0035</u> | 0.0079 | <u>0.0032</u> | **0.0052** | **0.0028** |
| sitting_rpy | 0.0481 | 0.0376 | <u>0.0384</u> | <u>0.0221</u> | **0.0268** | **0.0126** |
| walking_xyz | <u>0.0181</u> | <u>0.0105</u> | 0.0212 | 0.0111 | **0.0147** | **0.0068** |
| walking_half | <u>0.0292</u> | 0.0144 | 0.0301 | <u>0.0135</u> | **0.0245** | **0.0107** |
| walking_static | <u>0.0079</u> | <u>0.0034</u> | 0.0080 | 0.0035 | **0.0063** | **0.0026** |
| walking_rpy | <u>0.0962</u> | <u>0.0625</u> | 0.1457 | 0.0701 | **0.0702** | **0.0514** |

In Table 4, Y-SLAM refers to the direct elimination of feature points within the dynamic object frame by YOLOv5 target detection; YK-SLAM is the combination of YOLOv5 and improved K-means clustering to eliminate feature points within the dynamic object; YKP-SLAM is the proposed algorithm.

The comparison between Y-SLAM and YK-SLAM shows that the performance of YK-SLAM is better than Y-SLAM in the low dynamic environment, which is due to the fact that the number of dynamic points is smaller in the low dynamic environment. In contrast, Y-SLAM eliminates all the points in the dynamic object frame and deletes some static points by mistake, resulting in a reduction in constraints in the pose calculation, thus causing a decrease in pose accuracy. The performance of Y-SLAM is better than that of YK-SLAM in the high dynamic environment, which is due to the higher number of dynamic points and larger dynamic amplitude in the high dynamic environment. The area of the dynamic object frame is larger than that of the dynamic object, which allows Y-SLAM to reject more dynamic points and thus make its pose accuracy more accurate. YKP-SLAM with the addition of the probability update strategy achieves the best results in both low and high dynamic scenes. This is due to the fact that the probability update strategy assigns appropriate static probabilities to static and dynamic points and then adds all points to the pose calculation, which does not lead to either false deletion of static points or missed detection of dynamic points.

### 4.4. Real-Time Analysis

Real-time performance is one of the important evaluation indicators of SLAM systems. As shown in Table 5, in order to measure the real-time performance of the YKP-SLAM algorithm proposed in this paper, we test each module of the YKP-SLAM algorithm and the ORBSALM2 algorithm, respectively, under the highly dynamic "walking_xyz" sequence. In the table, A represents the YOLOv5 target detection module, B represents the ORB feature extraction module, C represents the improved K-means clustering module, D represents the probability update module, and E represents the normal tracking calculation pose module. Among them, the YOLOv5 target detection module and the ORB feature extraction module in the YKP-SLAM algorithm are run in parallel. The results show that the YOLOv5 target detection module cost less time than the ORB feature extraction module; that is to say, there is no need to wait for the detection results of YOLOv5 after the ORB feature extraction is completed. Therefore, in the case of sufficient computing power, adding the YOLOv5 module will not increase the system time. The average total time per frame of ORBSLAM2 and YKP-SLAM is 48.20ms and 62.05ms, respectively; that is, the running speed reaches 20

Fps and 16 Fps, respectively. Overall, YKP-SLAM basically meets the real-time performance of SLAM while ensuring accuracy in dynamic environments.

**Table 5.** The average running time of each module.

| Algorithm | A/ms | B/ms | C/ms | D/ms | E/ms | Total Time/ms |
|-----------|------|------|------|------|------|---------------|
| ORBSLAM2  | /    | 19.28 | /   | /    | 28.92 | 48.20        |
| YKP-SLAM  | 15.46 | 19.28 | 7.33 | 6.52 | 28.92 | 62.05       |

## 5. Conclusions

In this paper, a YKP-SLAM algorithm in dynamic environment is proposed. The algorithm first segments the whole current frame image by YOLOv5 target detection algorithm and improved K-means clustering algorithm and assigns a priori static probability to each feature point according to the segmentation result. The a priori static probability is used as the weight to calculate the initial camera pose, and then, the static probability of the feature points is updated according to the motion constraint and the epipolar constraint to solve the final camera pose. The algorithm in this paper is verified under the TUM dataset. Compared with the ORBSLAM2 algorithm, the accuracy and robustness of this algorithm are greatly improved in both low and high dynamic scenes. Compared with the other SLAM algorithms in dynamic scenes, the YKP-SLAM algorithm also achieves almost the best localization accuracy. In future work, we will propose a dense semantic map construction method in dynamic scenes based on the existing one and make full use of the advantages of localization accuracy in high dynamic scenes and the semantic information provided by YOLOv5 to realize path planning and obstacle avoidance in dynamic scenes.

**Author Contributions:** Conceptualization, L.L. and J.G.; methodology, J.G.; software, J.G.; validation, L.L., J.G. and R.Z.; formal analysis, L.L.; investigation, R.Z.; resources, J.G.; data curation, J.G.; writing—original draft preparation, L.L. and J.G.; writing—review and editing, J.G.; visualization, R.Z.; supervision, L.L.; project administration, L.L.; funding acquisition, L.L. All authors have read and agreed to the published version of the manuscript.

**Funding:** This research was supported in part by the Natural Science Foundation of Fujian Province under Grant 2022H6005 and 2022J01952, in part by the Initial Scientific Research Fund of FJUT under Grant GY-Z12079, Grant GY-Z21036, and Grant GY-Z20067.

**Data Availability Statement:** Not applicable.

**Acknowledgments:** The authors are grateful to the editors and the anonymous reviewers for their insightful comments and suggestions.

**Conflicts of Interest:** The authors declare no conflict of interest.

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
