# Peer review of "YKP-SLAM: A Visual SLAM Based on Static Probability Update Strategy for Dynamic Environments"

_electronics, doi:10.3390/electronics11182872_

Round 1
Reviewer 1 Report
This paper present a novel RGB-D SLAM approach for dynamic environments.
It presents a YOLO-based dynamic object detection, an improved K-means clustering, and a probability update method.
The paper is technically sound and well written.
There are some suggestions to the author.
- Real-time problem.
In line 177, you said that YOLOv5s is selected because of the real-time nature of the SLAM system.
By using the method author proposed, i think it will take a lot longer time because of the additional steps.
However, in the ablation study, the analysis of the operating time is lacking.
- YOLO misdetection problem.
The algorithm seems to depend entirely on the detection performance of YOLO. It would be good to analyze how the proposed method responds if there is no YOLO result for a certain period of time.
- geuristic
A new k-means clustering method was proposed to eliminate the heuristic(ness) of K-means clustering, but an excessively heuristic value of 20 to 40 cm was proposed. These values may vary depending on the depth accuracy of the sensor, etc., and a problem may occur if a dynamic object is close to the background.
- Confused expression
In line 215, "Number of pixels greater than 0.2M" is said by the author. But why?
In line 222, "much larger than the static background area" but there is any evidence for that.
- Weight update
In overall methods, the weights are only updated to be smaller. If the weights are updated continuously, The features which are tracked longer might have small weights. Is that true? or is there any method to compensate for that problem?
- Figure
In figure 8, why results from YKP-SLAM used a Sim(3) which contains scale correction?
It is a very major issue in comparison.
In figure 9, the legend box occludes the title of the figure.
Reviewer 2 Report
The work is clearly presented, and the contributions are clearly identified. Experiments seem reasonable and the results are clearly presented in general.
I have some concerns with a few equations. For example, in equation 3, the intrinsic parametric matrix (K) is inverted. As far as I understood, the authors are minimizing the reprojection error. Therefore, to compute the projection of 3D point xa, we should compute M xa = K [R | t] xa (not K-1)
In addition, I find the use of the T symbol to identify the pose a little confusing. The typical formulation will imply that T is a 4x4 matrix with the structure [R | t; 0 | 1]. I think that an explicit description of which is the structure of T would help the reader. Also, the meaning of the subindex S when computing the squared reprojection error should be clarified.
I also have a problem with equation 5. Even though we may know the camera pose and its internal parameters, the only information we can recover from an image point is the direction of the projecting ray. To know the position of the 3D point in this ray, we need additional information (for example the image of the 3D point from a different view). In this equation the use of T and K is confusing.
Equation 15 is the statement that pixel [x,y] belongs to the line [A, B, C], and I think that this equation is not necessary. Regarding to equation 16, the epipolar line corresponding to pixel [u1, v1]T is the product of the fundamental matrix (F) and the homogeneous coordinates of the pixel, therefore K matrix should not appear (unless [u1, v1]T are normalized image coordinates)
To finish, I don’t know where the values stated in line 384 come from.
Titles and numbers in figures 7 to 10 are difficult to read.
The problems associated with the K-means algorithm (lines 198-199) should be rewritten properly (the sentence is ill-formed).
One final question regarding the value of the segmentation threshold. This value is between 20 and 40cm as stated in the paper, what happens if the dynamic object has a bigger dimension along the direction of the optical axis?
Author Response
请参阅附件

Round 2
Reviewer 1 Report
The authors worte more appropriate results and discussions.
I think they correstly reflected the reviewers' comments.
In my opinion, this paper can be published in this form.
Thank you for your contribution to the community.